# Suicide Prevention in Nigeria: Can Community Pharmacists Have a Role?

**DOI:** 10.3390/pharmacy10050109

**Published:** 2022-09-02

**Authors:** Somto Chike-Obuekwe, Nicola J. Gray, Hayley C. Gorton

**Affiliations:** Department of Pharmacy, School of Applied Sciences, University of Huddersfield, Huddersfield HD1 3DH, UK

**Keywords:** mental health, suicide, suicide prevention, pharmacy, community pharmacists, low-middle income country, Nigeria

## Abstract

Suicide is a global public health problem and is among the leading causes of death worldwide. Over 700,000 people die by suicide globally each year, affecting all ages, genders, and regions. Community pharmacists are easily accessible and trusted frontline healthcare professionals. They provide pharmaceutical care to the community, yet their role is still yet to be fully optimised. With the expanding role of community pharmacists and their constant accessibility to the local population, they could have a potential role in suicide prevention and awareness in Nigeria through restriction of means, signposting to services, and conversations with patients built on trusting relationships. In this commentary, we review the literature on the involvement of community pharmacists in suicide prevention. In addition, we discuss the potential role of community pharmacists in Nigeria through establishing trusting relationships with patients, clinical counselling, and medication gatekeeping, given the existing gaps in knowledge and awareness of suicide prevention within community settings. This commentary also outlines potential barriers and solutions, making suggestions for future research.

## 1. Introduction 

Suicide is a global public health problem and is among the leading causes of death worldwide. Over 700,000 people die by suicide globally each year, affecting all ages, genders, and regions [1]. It is the fourth leading cause of death among 15–19 year olds after road accidents, tuberculosis, and interpersonal violence [1]. The World Health Organization (WHO) has made suicide reduction a priority, with the goal of reaching a one-third global reduction by 2030 [1]. However, low and middle-income countries (LMICs), including Nigeria, account for 77% of global suicide mortality, with limited resources for its prevention [2]. The current WHO suicide worldwide data reported an estimate of over 7000 suicides in Nigeria in 2019 [2]. However, the number of suicides is based on assumptions due to underreporting because of the stigma and illegality associated with suicide and the poor vital registration system in Nigeria’s healthcare system, and these factors make it difficult to accurately measure the burden of suicide in Nigeria [2,3].

Most community pharmacies in Nigeria are independently owned and operated by licensed pharmacists [4]. Community pharmacists are easily accessible and trusted frontline healthcare professionals who provide pharmaceutical care to the community [4]. Moreover, it costs less to receive health services in community pharmacies than in Nigerian hospitals due to free consultation and counselling [5]. This makes them the first point of contact for many people, and possibly the only healthcare system that most individuals access [5,6]. Various pharmaceutical services such as disease management, sex education, and immunisation offered by community pharmacies place them in the position to encounter different individuals and provide opportunities to recognise high-risk patients and offer suicide intervention and prevention through interactions and medication management [5,7]. The community pharmacist’s role is pivotal as medications are among the common methods of disease management and suicide [8]. Although community pharmacists have been reinventing themselves over the years to be more patient-oriented in their day-to-day activities, little is known about their knowledge and involvement in suicide prevention in Nigeria [6]. However, the exploration of the potential contribution of community pharmacists in suicide prevention strategies has begun in high-income countries (HICs), providing insights into community pharmacists’ interests, attitudes, and experiences [8,9,10,11,12]. This identifies the need for community pharmacists to be educated on suicide prevention to increase pharmacists’ confidence and knowledge on when and where to refer at-risk patients [7].

## 2. Community Pharmacists’ Current Involvement in Suicide Prevention 

The involvement of community pharmacists globally in suicide prevention was first mentioned 50 years ago by Gibson and Lott in the context that community pharmacists have a role in controlling medication overdose [13]. However, community pharmacy involvement was not explored until a few years ago. For example, Murphy et al.’s 2017 scoping review demonstrated the need for community pharmacists in suicide prevention but stated there is limited evidence of community pharmacists’ role and impact [14]. Furthermore, a thematic analysis of responses to open-ended survey questions exploring Canadian and Australian pharmacists’ experiences reported that pharmacists have a role in caring for and supporting individuals at high risk of suicide and those with suicidal thoughts, mainly through referral and triage [8]. A study conducted in Japan indicated a positive attitude towards suicide prevention from pharmacists who have received suicide prevention training [15]. 

In the United Kingdom, Gorton et al. gave a more detailed view of community pharmacy staff involvement in suicide prevention through an in-depth, open-ended qualitative interview study [12]. The community pharmacy staff saw their easy accessibility and relationships with patients as critical facilitators to suicide prevention efforts, and participants had positive attitudes toward discussing suicide; however, some barriers such as participants’ own beliefs, attitudes, lack of personal experience, and potential stigma were identified as hindering factors to suicide prevention efforts [12]. 

Gorton et al. [12] also identified referral and triage as essential in suicide prevention efforts within the community pharmacy, consistent with Murphy et al. [8]. The two studies identified that pharmacists and their teams collaborated with general practitioners (GPs) and non-governmental and community-based supports in referring those with suicidal ideation [8,12]. This supports a comprehensive Japanese suicide prevention programme based on the importance of broad multidisciplinary collaboration; however, it was not noted if community pharmacists are involved in this programme [16,17]. Currently, Washington State in the USA and Scotland are among the places that offer mandatory suicide prevention training for all licensed pharmacists and those employed by the health service, respectively [18,19]. Other countries, such as the UK and Australia, offer optional suicide prevention training for all healthcare providers, including pharmacists [20,21].

## 3. Community Pharmacists’ Potential Role in Suicide Prevention

Mental health in Nigeria remains grossly underfunded, with only seven government-owned psychiatry facilities and less than 150 psychiatrists serving a population of over 200 million people [22]. The few mental healthcare facilities are in the urban areas, making it difficult for most people to access mental healthcare and adding to the out-of-pocket health expenses that most people cannot afford [22]. The shortage of psychiatrists observed in the health system could be a challenge in meeting the mental healthcare need of the population, which is a risk factor for suicide. As of 2018, there were over 3000 registered community pharmacists in Nigeria [23]. Community pharmacists could develop their role and fill the apparent mental health workforce gap in the country. Indeed, a recent systematic review in Sub-Saharan Africa reported task shifting between health care professionals as an essential approach for improving access to mental health interventions [24].

Notwithstanding the differences in community pharmacy practices between countries, there are also similarities across global practices, such as the knowledge of drugs and the pharmacist–patient relationship [7,12]. Nigeria could learn from existing literature on suicide prevention involving community pharmacists in HICs, including the WHO suicide prevention guidelines, but this should be contextualised and adjusted to their pharmacy practice [1,9,11,12,14]. As stated previously, community pharmacists could have a significant role in suicide prevention through pharmacist–patient interaction and medication management, which is classified as restricting access to means [7]. Furthermore, community pharmacists could be a link to other healthcare systems in terms of referral and triage [12,25]. 

### 3.1. Community Pharmacists’ Involvement in Suicide Prevention through Restricting Access to Means

The WHO “Live Life” implementation guide to suicide prevention identified limiting access to the means of suicide as an effective intervention [1]. This could be achieved by banning hazardous substances, including medication and pesticides [1]. In some countries, restrictions to access to means have been shown to lower suicide rates. For example, Hawton et al.’s UK study reported the positive impact on suicide reduction over the years through the control of the paracetamol pack size that was implemented in 1998, reducing the pack size of paracetamol to 32 and 16 tablets when sold with or without a pharmacist present, respectively [26]. In addition, Corcoran et al. reported a significant 84% decrease in deliberate overdose of Distalgesic™ (paracetamol and dextropropoxyphene) in Ireland following its withdrawal from the retail markets [27]. Another study in Sri Lanka reported that limiting access to hazardous pesticides showed effectiveness in suicide reduction through implementing regulations and policies [28]. A similar process was followed in the restriction of purchases of the insecticide named Sniper™ in Nigeria, due to its involvement in suicide poisonings [29]; however, there is no evidence to report if this resulted in the reduction of suicide.

Several works of literature identify a common method of suicide in Nigeria to be hanging or jumping from high places; other means include self-poisoning with pesticides and medications [30,31,32]. There is an apparent connection between pharmacists and self-poisoning as their core duties are intrinsically linked to medication supply [33]. Based on the current practice in Nigeria, community pharmacists can supply both non-prescription and prescribed medications without prescriptions [34,35]. Although this is unethical, most patients still access their medications and related medication services through community pharmacies [34,35]. In a study in Benue State, Nigeria, community pharmacists admitted to supplying prescription-only medicines with or without a prescription due to competition with other retail pharmacies and patent medicine vendors. However, they expressed their awareness of the effects of drug misuse and drug abuse [35].

Community pharmacies in Nigeria could have a significant role in restricting access to means through medication management because they are medication experts [34]. This could be achieved by controlling the quantity of medication by managing refills for patients with mental health conditions, preventing stockpiling, maintaining sufficient records of patient’s medication histories, and providing adequate counselling on drug misuse and abuse [14,35,36]. Therefore, community pharmacists in Nigeria are positioned to intervene in suicide prevention, but lack of knowledge and awareness of suicide prevention and competition from other providers hinders them from seeing their potential role as medication gatekeepers.

### 3.2. Community Pharmacists’ Involvement in Suicide Prevention through Patient Communication

Pharmacists can be approached for counselling both in person and through phone calls without any need for an appointment [33]. Counselling is a vital pharmacy tool in improving therapeutic outcomes—this goes beyond pharmacotherapy interventions, including pharmacist–patient conversations and developing a trusting relationship [37].

Community pharmacists communicate with patients daily due to their easy accessibility; through these interactions, community pharmacists could help to identify suicide warning signs and those at risk who need a referral [7,33]. However, to optimise the full potential of their role in suicide prevention, community pharmacists need to establish a strong relationship and empathy for their patients. This helps to create a safe environment for communication [38].

Studies by Gorton et al. and Carpenter et al. identified established relationships as an opportunity for community pharmacists to intervene and prevent suicide [9,12]. They also perceived community pharmacy as a safe environment to talk about suicide [9,12]. Another study highlighted an effect on pharmacy students’ confidence in how they discussed suicide because of the established relationship with their patients [38]. 

There are no current reports of community pharmacists’ impact on suicide reduction to the best of our knowledge; however, effective communication has shown to improve patients’ health outcomes [39]. For example, a randomised clinical trial on the impact of pharmacists led education and counselling on epileptic patients showed an increase in the patient’s self-esteem and reduction in depression and anxiety experienced with epilepsy, which are suicide risk factors [40]. This was achieved through receiving education, counselling, and follow-up appointments with their clinical pharmacists [40]. 

## 4. Barriers to Community Pharmacists’ Involvement in Suicide Prevention

Despite positive existing literature on the potential role of community pharmacists in suicide prevention in Nigeria, significant barriers exist. These barriers include the following factors.

### 4.1. Stigma

The level of stigma in Nigeria is high, as suicide is criminalised under Section 327 of the criminal code [41]. This could be a barrier to its inclusion in pharmacy schools’ curricula, preventing a start to creating awareness and intervention toward suicide. If a community-based suicide prevention strategy is encouraged, community pharmacists could assist in reducing suicide stigma and patient hospitalisation in Nigeria.

### 4.2. Lack of Multisectoral Collaboration

Multisectoral collaboration is an effective part of suicide prevention in terms of modifying a patient’s treatment plan and referrals. Many studies have recommended collaboration; however, in LMICs, most healthcare professionals prefer to work without pharmacists [14,42]. Effective interactions between community pharmacists and other healthcare providers could be essential. Most community pharmacists in Nigeria do not have access to their patients’ medication records [43]. The lack of access to patient data could be a significant issue in providing effective suicide intervention and referral.

### 4.3. Lack of Referral Service

Community pharmacy could be a link between suicidal patients and other healthcare services [44]. Referral and triage have been highlighted as barriers to effective suicide prevention because of the lack of collaboration between community pharmacists and other healthcare professionals [12]. For example, participants in Gorton et al.’s study expressed a dilemma regarding where to refer suicidal patients they encounter. They mentioned referring most patients to general practitioners or their own family members [12]. An established referral system has been highlighted as a critical path of effective suicide prevention [12].

### 4.4. Lack of Education and Training

Most pharmacy schools in Nigeria do not include suicide prevention training in their curriculum [45]. Even the mandatory continuing education programme recommended and adopted in Nigerian pharmacy practice to update pharmacists’ knowledge and professional skills lacks training on mental healthcare, including suicide prevention [45]. Lack of adequate training could influence how community pharmacists interact with people about suicide. Studies have reported low confidence in interacting with those at risk based on how to communicate, the right language, and where to refer patients [9,12,25,38].

### 4.5. Lack of Reimbursement for Suicide Risk Assessment

Without commensurate financial incentives, community pharmacists may not be motivated to provide suicide risk assessment services and have limited time to provide such services [25].

## 5. Facilitators for Community Pharmacists’ Involvement in Suicide Prevention

The facilitators for involvement mirror the barriers, and here, we discuss two major facilitators in more detail.

### 5.1. Education and Training

Education and training are needed for community pharmacists to be able to communicate and effectively intervene with individuals with suicidal ideation as their lack of confidence, lack of training, and education has been a reoccurring barrier [7,9,11,12].

Suicide prevention needs to be added to pharmacy schools’ curricula and pharmacists’ continuing education programmes to evolve the pharmacist’s role in suicide prevention.

A Washington State survey assessing pharmacy students’ confidence, knowledge, and skills in suicide prevention pre- and post-training reported their increased confidence and skills in recognising suicide warning signs and their ability to have an emotional connection and intervene [38], supporting all the studies that have been identified in this commentary [7,11,12,25,33].

### 5.2. Provision of Incentives 

For community pharmacists to include suicide risk assessment as a sustainable part of their pharmaceutical services, supportive policy action is needed to offer incentives for establishing a suicide risk assessment programme, including its continuous maintenance.

## 6. Discussion and Conclusions

Community pharmacists’ roles in suicide prevention have been grouped into two intervention routes: restricting access to means and pharmacist–patient interactions [12]. Restriction of means is an obvious role for community pharmacists, as they are medication gatekeepers and experts. Studies have identified established patient relationships as facilitators in suicide prevention activities [8,13]. The studies highlighted the community pharmacy environment as a safe place where people with suicide ideation could converse with the pharmacy teams about their feelings and emotions [12]. In community pharmacy settings in Nigeria, there is a need to develop private consultation areas within the pharmacy as safe places for those at risk of suicide. The existing reports in some HICs on community pharmacists’ encounters with suicidal patients or those with suicidal ideation could indicate the importance of the availability of data in establishing a suicide prevention programme. Furthermore, there is an urgent need for adequate reporting of suicide in Nigeria by acting on stigma and legislation. This could create awareness and possible government interest in prioritising suicide prevention programmes.

As more literature, policies, and reports are published, the role of community pharmacists in suicide prevention will be increasingly recognised. Although all the current but limited research focuses on HICs, this could be a useful starting point. There is a need for a better understanding of Nigeria’s suicide prevention strategy. Hence, more in-depth research on suicide and community pharmacists’ knowledge, attitudes, and perceptions towards suicide need to be conducted in Nigeria to fully understand the best way to educate and train community pharmacists for a future role in suicide risk assessment and prevention.

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
