# Peer review of "Suicide Prevention in Nigeria: Can Community Pharmacists Have a Role?"

_pharmacy, 2022, doi:10.3390/pharmacy10050109_

Round 1
Reviewer 1 Report
The manuscript submitted has the potential to provide community pharmacists with guidance as to the potential roles pharmacists can have in suicide prevention; however, the manuscript submitted needs to be significantly reworked to provide this guidance. Currently the manuscript flips between prevention, risk management, and educational initiatives making it difficult to follow. It would be suggested that the authors focus on the need for suicide prevention supports and the supports community pharmacy could offer to address this unmet need (suicide prevention). Lastly there are a number of statements that would require referencing throughout the manuscript (e.g., 38, 73, 112, etc.)
Author Response
Authors’ Response to Reviewers’ Comments
Reviewer 1
- The manuscript submitted has the potential to provide community pharmacists with guidance as to the potential roles pharmacists can have in suicide prevention; however, the manuscript submitted needs to be significantly reworked to provide this guidance.
Authors’ response: Thank you for acknowledging that the manuscript has potential in developing the pharmacy workforce and enhancing the knowledge on suicide prevention. Thank you also for thoroughly reviewing the manuscript.
- Currently the manuscript flips between prevention, risk management, and educational initiatives making it difficult to follow. It would be suggested that the authors focus on the need for suicide prevention supports and the supports community pharmacy could offer to address this unmet need (suicide prevention). Lastly there are a number of statements that would require referencing throughout the manuscript (e.g., 38, 73, 112, etc.)
Authors’ response: We agree with your helpful suggestions on the arrangement of the manuscript and its direction. We have edited the manuscript to ensure that explanation is straight forward. We move from giving some introduction of suicide epidemiology and community pharmacy practise in Nigeria (line 2) into the introductory context of community pharmacists’ current role in suicide prevention (Line 56), their potential role (Line 88) through medication management (Line 111) and patient-pharmacist interaction (Line 152). Furthermore, we discussed the potential barriers (Line 177), and facilitators (Line 222) in incorporating suicide prevention in community pharmacists.
We have also revised the manuscript to ensure consistency in the referencing using the Zotero reference tool. (Line 51-52 and Line 106) (References 1, 8,9, 10, 11,12,14).
Reviewer 2 Report
This reviewer read this manuscript with great interest, since this reviewer does not know the medical situation in Africa, including Nigeria. This reviewer generally agrees with the author's assertions regarding the sociopsychological aspects. This reviewer has great expectations that this manuscript probably contributes to the research and social contribution activities in Africa. This reviewer provides several suggestions and comments, to format this manuscript as a scholarly article and increasing impacts.
Major comments
1) It has been established to be important that the cooperation among various occupations in their respective areas of expertise develop the comprehensive suicide prevention programme. In this consensus, it is important for authors to clearly state the role of pharmacists in comprehensive suicide prevention programs. This reviewer understood that authors were saying the pharmacist can act as a gatekeeper. If this reviewer misunderstood, please describe more detailed statements in this manuscript. Comprehensive suicide prevention programs in Japan revealed the importance of multidisciplinary collaboration in enlightenment and developing gatekeepers for suicide prevention (PMID: 31795379, 32859665, 33806105 and 34475170). Furthermore, please add WHO reports and guidelines for suicide prevention.
2) Introduction section.
・Please clearly state the main purpose of this manuscript.
・Would it be possible to add not only international statistics but also Nigeria's suicide statistics (even if it is uncertain statistics)?
・Pleases add detailed medical statistics, especially standardised numbers of psychiatrists, GPs and pharmacists in Nigeria.
3) Please checked carefully the reference lists and must be corrected.
For example, the #3 could not be found in PubMed. “2020 Nov 25;1.” Is not correct, but “2022, 11(6):829-839.”?
Minor comments
1) Please provide the reference for “The exploration of the potential contribution of community pharmacists in suicide prevention strategies has begun in High-Income countries (HICs)” (L35-37)
2) “A study in Sri Lanka reported that limiting access to hazardous pesticides showed effectiveness in suicide reduction through implementing regulations and policies” (L155-157)
Success of restriction in Sri Lanka is very famous, whereas we know a number of reports in various countries regarding the restriction measures. Please add other references (ie, PMID: 28807587)
3) Please provide the reference for “The exploration of the potential contribution of community pharmacists in suicide prevention strategies has begun in High-Income countries (HICs)” (L35-37)
4) Please correct “20130” (L28)
Author Response
Authors’ Response to Reviewers’ Comments
Reviewer 2
Comments and Suggestions for Authors
This reviewer read this manuscript with great interest, since this reviewer does not know the medical situation in Africa, including Nigeria. This reviewer generally agrees with the author's assertions regarding the sociopsychological aspects. This reviewer has great expectations that this manuscript probably contributes to the research and social contribution activities in Africa. This reviewer provides several suggestions and comments, to format this manuscript as a scholarly article and increasing impacts.
Authors’ response: Thank you for your insightful suggestions and taking the time to review our manuscript.
Major comments
- It has been established to be important that the cooperation among various occupations in their respective areas of expertise develop the comprehensive suicide prevention programme. In this consensus, it is important for authors to clearly state the role of pharmacists in comprehensive suicide prevention programs.
Authors’ response:
Currently the role of community pharmacists in suicide prevention is still a new exploring field. Community pharmacists have not currently been included in most countries prevention policy including Nigeria. However, we have added a short summary on the role of community pharmacy in the wider system context, both in the abstract and the main body of the commentary (Line 111-175).
We also included the current countries that have included pharmacists into the suicide prevention training programme for a clear understanding on how new and evolving this role is (Line 83-86) (references 18,19,20 and 21)
- This reviewer understood that authors were saying the pharmacist can act as a gatekeeper. If this reviewer misunderstood, please describe more detailed statements in this manuscript. Comprehensive suicide prevention programs in Japan revealed the importance of multidisciplinary collaboration in enlightenment and developing gatekeepers for suicide prevention (PMID: 31795379, 32859665, 33806105 and 34475170). Furthermore, please add WHO reports and guidelines for suicide prevention.
Authors’ response: Yes, the reviewer is right. We are explaining that community pharmacists have a potential role in suicide prevention through medication gatekeeping and patient-pharmacist relationship-building and interaction, whilst highlighting the need of education and training for pharmacists to effectively intervene (Line 111-175) (References 22-40).
We appreciate the articles you recommended – they were very informative, and the author included it in the manuscript highlighting those pharmacists were not included but there is still need of multidisciplinary collaboration (Line 81-83) (References 16 and 17).
We have added the WHO ‘Live Life’ suicide prevention implementation guide and the 2019 suicide global health estimates (references 1 and 2)
- Introduction section.
・Please clearly state the main purpose of this manuscript.
・Would it be possible to add not only international statistics but also Nigeria's suicide statistics (even if it is uncertain statistics)?
・Pleases add detailed medical statistics, especially standardised numbers of psychiatrists, GPs and pharmacists in Nigeria.
Authors’ response: We have stated the main purpose of the manuscript in the abstract and the main body of the manuscript (Line 11-16). The role of community pharmacists in suicide prevention is a new field, additionally, suicide is still criminalised in Nigeria, which creates stigma in its reporting. This leads to the barrier of it being taught in pharmacy schools including carrying out research on the topic.
There are limited data on the community pharmacy role in suicide prevention in Africa, however, after a comprehensive literature search, we did find and included Nigeria statistics about the numbers of psychiatrists and community pharmacists, as the reviewer recommended (Line 29-33 and line 88-99) (References 2, 22,23 and 24)
- Please checked carefully the reference lists and must be corrected.
For example, the #3 could not be found in PubMed. “2020 Nov 25;1.” Is not correct, but “2022, 11(6):829-839.”?
Authors response: I have revised it using the Zotero reference tool. I can understand the confusing date- the day the manuscript was published online was on November 25, 2020, and the International Journal Health Policy Management accepted it on June 2022. Therefore, currently I referenced the date the journal published as that’s where I read the article (Line 297-299).
Minor comments
- Please provide the reference for “The exploration of the potential contribution of community pharmacists in suicide prevention strategies has begun in High-Income countries (HICs)” (L35-37)
Authors response: I have included references for High-income-countries after careful literature research. These references were included according to the pharmacist interaction, experiences, attitude and training in suicide prevention (Line 52) (References 8-12).
- A study in Sri Lanka reported that limiting access to hazardous pesticides showed effectiveness in suicide reduction through implementing regulations and policies” (L155-157). Success of restriction in Sri Lanka is very famous, whereas we know a number of reports in various countries regarding the restriction measures. Please add other references (ie, PMID: 28807587)
Authors response: I included another reference on the withdrawal of Distalgesic from Ireland retail market (Reference 27). This decision was based on my personal experience as a community pharmacist where I have encountered the high purchase of NSAIDs by patients. Additionally following a clear deliberation we decided to maintain the Sri-Lanka article in the manuscript as it also gives a mirrored view on the purchase of pesticide and insecticide in Nigeria (Reference 28).
- Please provide the reference for “The exploration of the potential contribution of community pharmacists in suicide prevention strategies has begun in High-Income countries (HICs)” (L35-37)
Authors response: I have included references for High-income-countries after careful literature research. These references were included according to the pharmacist interaction, experiences, attitude, and training in suicide prevention (Line 52) (References 8-12).
4) Please correct “20130” (L28)
Author’s response: This has been carefully adjusted after critical review (Line 27).
Round 2
Reviewer 2 Report
The authors made all the recommended corrections.
I feel satsfied with the novel version of the manuscript.